

# All you can eat: the functional response of the cold-water coral *Desmophyllum dianthus* feeding on krill and copepods

Juan Höfer[1,2], Humberto E. González[1,2], Jürgen Laudien[3], Gertraud M. Schmidt[3], Verena Häussermann[4,5] and Claudio Richter[3,6]

[1] Instituto de Ciencias Marinas y Limnológicas, Universidad Austral de Chile, Valdivia, Chile
[2] Centro FONDAP de Investigación en Dinámica de Ecosistemas Marinos de Altas Latitudes (IDEAL), Valdivia, Chile
[3] Alfred-Wegener-Institut, Helmholtz-Zentrum für Polar und Meeresforschung, Bremerhaven, Germany
[4] Facultad de Recursos Naturales, Escuela de Ciencias del Mar, Pontificia Universidad Católica de Valparaíso, Valparaíso, Chile
[5] Huinay Scientific Field Station, Huinay, Chile
[6] Fachbereich Biologie/Chemie, Universität Bremen, Bremen, Germany

## ABSTRACT

The feeding behavior of the cosmopolitan cold-water coral (CWC) *Desmophyllum dianthus* (Cnidaria: Scleractinia) is still poorly known. Its usual deep distribution restricts direct observations, and manipulative experiments are so far limited to prey that do not occur in CWC natural habitat. During a series of replicated incubations, we assessed the functional response of this coral feeding on a medium-sized copepod (*Calanoides patagoniensis*) and a large euphausiid (*Euphausia vallentini*). Corals showed a Type I functional response, where feeding rate increased linearly with prey abundance, as predicted for a tentaculate passive suspension feeder. No significant differences in feeding were found between prey items, and corals were able to attain a maximum feeding rate of 10.99 mg C h$^{-1}$, which represents an ingestion of the 11.4% of the coral carbon biomass per hour. These findings suggest that *D. dianthus* is a generalist zooplankton predator capable of exploiting dense aggregations of zooplankton over a wide prey size-range.

## INTRODUCTION

In recent years, cold-water corals (CWC) have received increasing attention from the scientific community as they were considered particularly vulnerable to global warming and ocean acidification (*Doney et al., 2009*; *Maier et al., 2012*; *Jantzen et al., 2013a*, *2013b*; *Lebrato et al., 2016*). Although there are recent findings on how CWC may response to global change (*Maier et al., 2012*, *2013*; *Jantzen et al., 2013b*; *McCulloch et al., 2012a*, *2012b*; *Findlay et al., 2014*; *Gori et al., 2016*), there are still several uncertainties about their adaptive capacity. Part of them are due to the paucity of basic knowledge of the biology of CWC species, particularly in terms of growth, life cycle, and feeding. For example,

Corresponding author
Juan Höfer, juanhofer@gmail.com

several CWCs seem able to up-regulate the pH of their internal calcifying fluid in order to cope with lower seawater pH (*McCulloch et al., 2012a*). However, the capacity to withstand unfavorable conditions likely depends on the nutrition level and the overall fitness of the corals (*Guinotte et al., 2006*; *Jantzen et al., 2013a*).

*Desmophyllum dianthus* (Esper, 1794) is a cosmopolitan CWC species (*Cairns, 1983*; *Cairns, Försterra & Häussermann, 2005*), that forms solitary polyps of up to 40 cm in height and 6.3 cm in diameter (*Försterra & Häussermann, 2003*), and usually lives beyond the reach of divers between 35 and 2,460 m depth (*Cairns, 1995*). In the northern hemisphere, it is often found in deep-water coral communities (*Freiwald et al., 2004*; *Roberts et al., 2009*) associated with *D. pertusum* (Linnaeus, 1758) and *Madrepora oculata* Linnaeus, 1758 (*Reveillaud et al., 2008*; *Heindel et al., 2010*). In the southern hemisphere it constitutes the main CWC species in shallow coral banks off New Zealand and Chile (*Squires, 1965*; *Cairns & Stanley, 1982*). However, the geographical distribution of *D. dianthus* may probably shrink in the near future due to physiological stress caused by ocean acidification and especially global warming (*Gori et al., 2016*).

In the Comau Fjord (northern Patagonia, Chile; 42° 22.767 S, 72° 25.534 W) *D. dianthus* is abundant (*Försterra & Häussermann, 2003*; *Jantzen et al., 2013a*, *2013b*; *Fillinger & Richter, 2013*) and locally dominates the benthic hard-bottom community (*Försterra & Häussermann, 2003*; *Cairns, Försterra & Häussermann, 2005*). Here, the coral banks attain abundances of up to 1,500 coral individuals per square meter (*Cairns, Försterra & Häussermann, 2005*) and single individuals have been found as shallow as seven m (*Försterra et al., 2005*). A surface layer of brackish water limits the upper coral distribution around 12 m depth (*Försterra & Häussermann, 2003*; *Cairns, Försterra & Häussermann, 2005*), while in the deep basin of the fjord the reduction of its abundance seems to be related to higher metabolic costs due to lower oxygen concentration combined with lower pH (*Fillinger & Richter, 2013*). This shallow upper distribution in the Comau Fjord represents a unique opportunity to collect *D. dianthus* for manipulative experiments by scientific SCUBA diving.

In the past, the role of zooplankton in the diet of CWC has been questioned due to the low abundance of zooplankton in deep CWC habitats (*Kiriakoulakis et al., 2005*) and an insufficient number of nematocysts to properly capture zooplankton (*Lasker, 1981*). However, recent findings suggest that zooplankton may play an important role in the diet of *D. dianthus* (*Carlier et al., 2009*; *Mayr et al., 2011*) being a major energy source for this CWC species (*Naumann et al., 2011*). The few direct estimates of *Desmophyllum* feeding rates available are based on laboratory experiments with non-naturally occurring prey, that is, *Artemia salina* (Linnaeus, 1758) (*Tsounis et al., 2010*), thus, the feeding behavior of this CWC on natural zooplankton is virtually unexplored.

The functional response examines predator–prey interactions that have direct implications for population dynamics of both prey and predator (*Holling, 1965*; *Murdoch, 1977*), and constitutes a key concept in trophic ecology (*Holling, 1959*, *1966*), as it describes a species capability to exploit resources for survival, growth, and recruitment, which in the case of corals also affects their mortality risk due to environmental stress

(*Anthony et al., 2009*). The main goal of this study was to examine the natural feeding behavior of the CWC *D. dianthus*. In the Mediterranean Sea, stable isotopes and observations suggest that copepods and euphausids may be important prey items for *Desmophyllum* (*Carlier et al., 2009*; *Tsounis et al., 2010*). Our own observations show that this seems to be the case also for *D. dianthus* in the Comau Fjord. We therefore quantified the functional response of Patagonian *D. dianthus* feeding on two corresponding natural zooplankters: *Calanoides patagoniensis* Brady, 1883, a medium-sized copepod, and *Euphausia vallentini* Stebbing, 1900, a large euphausiid. The comparision between the functional responses on each prey allows us to attain a first and comprehensive view of *D. dianthus* natural feeding behavior.

## MATERIALS AND METHODS

### *D. dianthus* sampling

*D. dianthus* was collected by scientific SCUBA divers in the Comau Fjord (Northern Patagonia, Chile; see details in *Försterra et al., 2005*; *Fillinger & Richter, 2013*) during September and December 2016 and May 2017 (i.e., spring, summer, and fall season). The collection of corals was approved by the sub-secretariat of fisheries and farming within the Chilean Ministry of Economy, Development & Tourism (file number: 1760). In each season circa twenty specimens of similar size (3–4 cm long, 1.5–2 cm diameter) were individually chiseled off highly populated rocks near 42° 14.935 S, 72° 30.880 W at 18–20 m depth. Corals with polyp tissue almost down to their basis were selected to avoid infestations by endolithic or boring sponges (*Försterra et al., 2005*; *Jantzen et al., 2013b*). Immediately after their collection, corals were placed in a plastic container filled with their ambient seawater for transportation. This prevented corals from suffering an osmotic shock when divers ascended through the shallow brackish surface layer present within this fjord (*Schwabe et al., 2006*). Corals were immediately transported to Huinay Scientific Field Station (HSFS), where they were placed in flow-through aquaria constantly supplied with unfiltered fjord water from 24 m water depth to resemble their natural conditions. Parts of the skeletons not covered with tissue were cut underwater with a diamond blade equipped rotary tool (DREMEL Europe, Breda, The Netherlands) before removing all epibionts (e.g., microbial mats, tube-dwelling polychaetes, bryozoans) and gluing the corals to polyethylene screws (Super Flex Glue Gel; UHU GmbH & Co KG, Bühl, Germany). Corals were allowed to acclimatize for at least 48 h before the incubation experiments started. During this period, corals were regularly checked and only healthy corals with extended tentacles during night hours were used for the experiments. Then, the functional response of corals feeding on krill and copepods was assessed (see details below and in Table 1).

### Zooplankton sampling

Zooplankton samples were collected in front of HSFS during sunset and the first night hours. Several vertical hauls from 100–150 m depth to the surface were made using a bongo net (mouth of 40 cm diameter) equipped with a non-filtering cod end to reduce zooplankton stress during sampling. For sampling medium-sized zooplankton

**Table 1 Details of incubations used to assess the functional response of *D. dianthus*.**

| Date | Prey type | Min PA | Max PA | N | Start time | End time | O$_2$ con |
|---|---|---|---|---|---|---|---|
| 11/09/2016 | EV | 1.26 | 11.39 | 9 | 00:30 | 03:30 | 8.8 |
| 13/09/2016[a] | CP | 2.53 | 22.78 | 6 | 23:30 | 02:30 | 1.7 |
| 15/12/2016 | EV | 1.26 | 10.12 | 8 | 01:00 | 04:00 | 5.4 |
| 17/12/2016 | CP | 5.06 | 25.31 | 9 | 23:00 | 02:00 | 2.2 |
| 06/05/2017[b] | EV | 1.26 | 11.39 | 8 | 23:30 | 02:30 | 3.2 |
| 14/05/2017[c] | CP | 6.33 | 50.63 | 8 | 23:30 | 02:30 | 4.1 |
| global | EV | 1.26 | 11.39 | 25 | | | 5.8 |
| global | CP | 2.53 | 50.63 | 23 | | | 2.7 |

| Organism | Mean carbon biomass (mg C) | SE |
|---|---|---|
| *Desmphyllum dianthus* | 96.198 | 8.5000 |
| *Euphausia vallentini* | 4.713 | 0.3300 |
| *Calanoides patagonensis* | 0.043 | 0.0063 |

**Notes:**
Date of the incubation, prey type, minimum prey abundance (Min PA, prey L$^{-1}$), maximum prey abundance (Max PA, prey L$^{-1}$), number of corals used (*N*), incubation starting time (Start time), incubation ending time (End time) and mean % of oxygen consumed during incubations (O$_2$ con). CP and EV stand for *Calanoides patagoniensis* and *Euphausia vallentini*, respectively.

[a] During this incubation two bottles were discarded before starting the functional response experiment due to unhealthy (i.e., non-swimming) copepods.

[b, c] The corals used for these incubations were collected on different days.

(i.e., *C. patagoniensis*) the bongo net was equipped with a 200 μm mesh size, whereas, 500 μm mesh size was used for larger zooplankton (i.e., *E. vallentini*). Immediately after sampling, the zooplankton was placed in a cooler filled with unfiltered seawater from 24 m depth and transported to HSFS. There, healthy individuals of the target species were identified and selected under a stereomicroscope. Selected individuals of *C. patagonenis* and *E. vallentini* were then placed in 500 mL Schott bottles (Schott AG, Mainz, Germany) filled with 790 mL of seawater filtered through a 20 μm sieve to remove all zooplankters. Bottles containing zooplankton were then placed in a dark room for 2 or 3 h inside tanks with a constant flow of seawater from 24 m depth, maintaining zooplankton at their environmental temperature while reducing their stress. All bottles were checked immediately before performing functional response incubations and bottles containing unhealthy zooplankton (i.e., non swimming) were discarded (see Table 1).

## Functional response incubations

Normally, eight or nine specimens of *D. dianthus* (see Table 1) were placed upside down, similar to their natural orientation, inside bottles containing healthy zooplankton (Fig. 1A), then moved from HSFS laboratory to HSFS jetty and deployed hanging from the jetty at ≈8 m depth. This experimental set up simulates in situ conditions of temperature and light. On different days (i.e., one for each prey type), corals experienced increasing zooplankton concentrations from 1.26 to 11.39 euphausiids L$^{-1}$ and from 2.53 to 50.63 copepods L$^{-1}$ (see incubation details in Table 1). Incubations started around midnight and lasted 3 h to avoid oxygen consumptions larger than 10%, which might alter coral feeding behavior during the experiments (Table 1). Initial and final

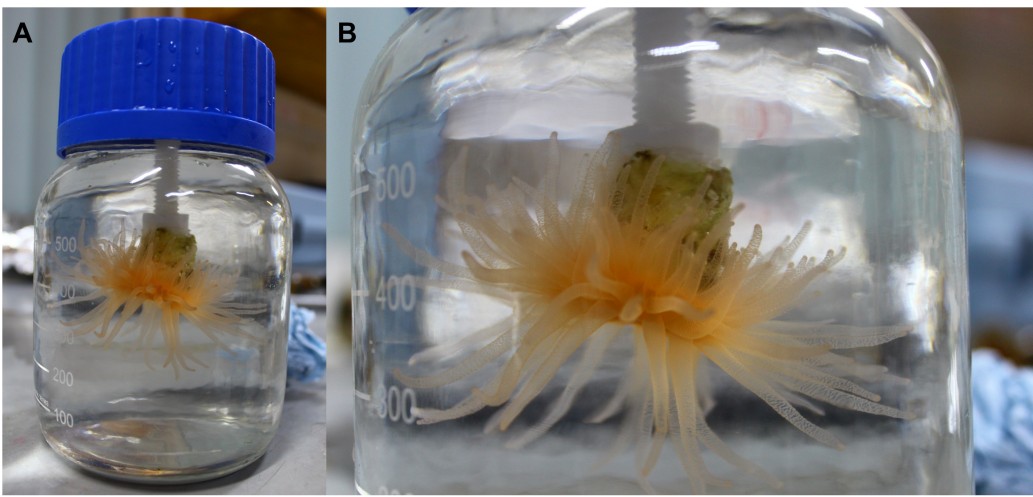

**Figure 1 Coral in the laboratory at the end of an incubation.** (A) Specimen of *D. dianthus* inside an incubation bottle. (B) Fully extended tentacles of the same coral in detail. Photo by N. García-Herrera.

oxygen concentrations were measured using a Hach HQ40D multiparameter probe. We checked if the tentacles of the corals were extended at the beginning and end of the incubations in order to discard any coral without totally extended tentacles, although no coral was discarded due to this reason. Back at the laboratory of HSFS, corals were returned to the acclimatization tanks. The remaining zooplankton (i.e., not captured by corals) was collected from the incubation bottles using a 20 µm sieve and then counted under a stereomicroscope. Zooplankton and corals were subsequently frozen at −20 °C and transported to the laboratory, where corals and zooplankton were retrieved and dried in an oven at 58 °C for 48 h. Euphausiids and copepods were ground to a fine powder with a mortar and pestle, while corals were previously decarbonated by immersion in liquid HCl (10%) at room temperature following the procedure also used for *Desmophyllum* samples by *Carlier et al. (2009)*. For elemental analysis, one coral (*D. dianthus*), one euphausiid (*E. vallentini*), and 20 copepods (*C. patagoniensis*) pooled together, were placed on pre-combusted glass fiber filters and assayed for carbon biomass content (Table 1). Samples were weighed into tin cups (3.3 × 5 mm), combusted at 900 °C, and analyzed in a continuous flow isotope ratio mass spectrometer (Flash EA200 IRMS Delta Series; Thermo Scientific, Bremen, Germany).

## Data analysis

Statistical analyses were performed using the program R version 3.1.0 (*R Core Team, 2012*). First of all, we analyzed if there were significant differences in the functional responses for each prey among different seasons. An analysis of covariance (ANCOVA) was performed using the *aov* function from the package *stats* to check if the slopes for each prey and season were significantly different. Since the slopes for *E. vallentini* ($n = 25$, $F = 0.002$, $p$-value = 0.97) and *C. patagoniensis* ($n = 23$, $F = 1.911$, $p$-value = 0.18) were not significantly different among seasons, all data for each prey were pooled together for subsequent analysis.

*Holling's (1959)* functional response describes three different types (i.e., I, II, and III) of relationship between predator feeding rate and the abundance of its prey. In short, Type I consists of a linear increase in the feeding rate with higher prey abundances, whereas Type II depicts a feeding rate that increases with a decelerating rate with higher prey abundances until the feeding rate reaches an asymptote (i.e., maximum feeding rate) that represents the saturation of the predator and directly depends on the prey handling time. Type III is similar to Type II, predator saturation with high prey abundances, but, with low prey abundances the increase in feeding rate is closer to an exponential fit than a linear one. This is the consequence of the predator learning time, that is, the improvement in the attack and handling efficiency of the predator feeding on that particular prey (see details in *Holling, 1959*).

Second-order logistic regressions were used to test which type of functional response was exhibited by *D. dianthus* preying on *C. patagoniensis* and *E. vallentini* (see details in *Trexler, McCulloch & Travis, 1988*; *Juliano, 2001*; *Alexander et al., 2012*). The effect of prey abundance (i.e., food availability) on the proportion (parts per unit) of prey ingested by corals was explored using the function *glm* from the *stats* package. Type I responses present non-significant linear terms (*Buckel & Stoner, 2000*); whereas Type II has a significantly negative first-order term and Type III responses show a significantly positive first-order term, followed by a significantly negative second-order term (*Juliano, 2001*).

According to functional response Type I, ingestion rate increases linearly with prey abundance. Model selection was used to test which kind of fit, linear or logarithmical, better explained the relationship between coral ingestion rate and prey abundance. Model selection was performed using second order Akaike Information Criteria (AICc). Functions *lm* (package *stats*) and *aictab* (package *AICcmodavg*) were used for linear regressions and model selection, respectively. *C. patagoniensis* and *E. vallentini* represent two different prey types due to their differences, for example, in size, swimming ability and feeding behavior. To examine if *D. dianthus* displayed different feeding responses related to the prey type, the relationship between coral daily ration (% of coral carbon biomass ingested $d^{-1}$) and prey biomass (mg C $L^{-1}$) was compared for each prey item. If the prey type does not affect the response of *D. dianthus*, the linear regressions for *C. patagoniensis* and *E. vallentini* should present similar slopes. The similarity of the slopes for *C. patagoniensis* and *E. vallentini* was tested by an ANCOVA performed using the *aov* function from the package *stats*.

## RESULTS

Krill and copepods were actively swimming inside the incubation bottles at the end of each experiment, while corals had their tentacles fully extended (Fig. 1B). Oxygen consumption during all incubations was always lower than the 10% of the initial oxygen concentration (Table 1). All these evidences support the reliability of the results obtained during the incubations.

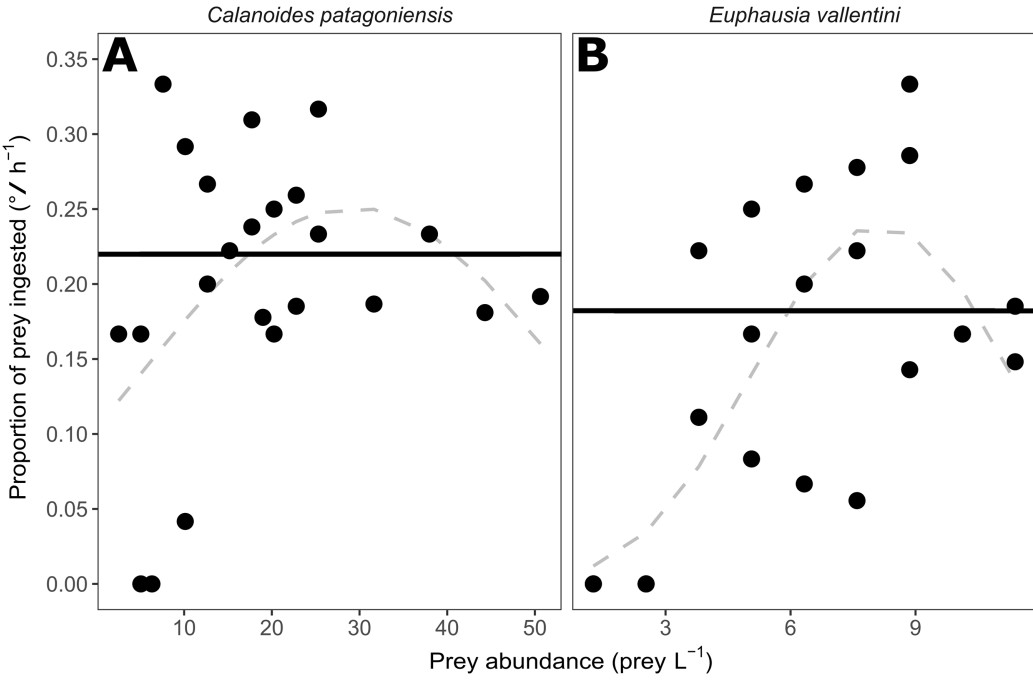

**Figure 2 Scatter plot showing the prey abundance (number of prey L⁻¹) and the proportion of prey (parts per unit) ingested by corals (°/ h⁻¹).** (A) Data for *C. patagoniensis* incubations. (B) Data for *E. vallentini* incubations. Solid black lines represent the mean proportion of prey ingested by feeding corals, while dashed gray lines correspond to the non-significant second-order logistic regression fits.

**Table 2 Functional response type.**

| Prey | First-order term | Second-order term |
|------|------------------|-------------------|
| *Calanoides patagoniensis* | 0.63 (0.81) | −1.08 (0.68) |
| *Euphausia vallentini* | 4.38 (0.34) | −3.11 (0.43) |

Note:
Terms and their *p*-value (inside brackets) for the second-order logistic regressions used to analyze the relationship between prey abundance and the proportion of prey ingested by corals (i.e., analysis to determine functional response type).

## Functional response type

The proportion of prey (parts per unit) ingested by *D. dianthus* showed no pattern over a wide range of prey abundances for *C. patagoniensis* and *E. vallentini* (Fig. 2). The second-order logistic regressions for *C. patagoniensis* and *E. vallentini* (Fig. 2) did not present any significant term (Table 2) pointing out that *D. dianthus* displayed a functional response Type I when feeding on both prey. Corals actually feeding ate a mean 21.99% and 18.22% of copepods and euphausiids, respectively (Fig. 2), with only a marginally significant difference between the percentage of krill and copepods eaten (Kruskal–Wallis, $\chi^2 = 2.88$, *p*-value = 0.09). The proportion of prey eaten was approximately constant and did not depend on the number of prey offered (Fig. 2), which means that coral ingestion rate would increase linearly with prey abundance (Fig. 3).

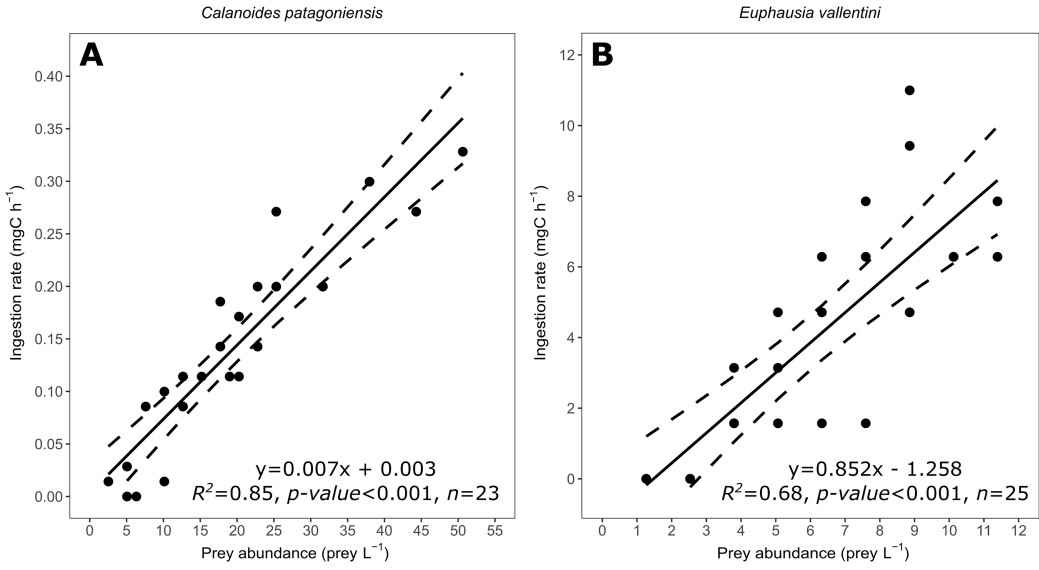

**Figure 3 Scatter plot showing the relationship between prey abundance (prey $L^{-1}$) and coral ingestion rate (mg C $h^{-1}$).** (A) Data for *C. patagoniensis*. (B) Data for *E. vallentini*. Solid lines represent linear fits, whereas dashed lines correspond to their 95% confidence intervals.

**Table 3 Model selection for the linear and logarithmic models fitted to coral ingestion rate and prey abundance for *C. patagoniensis* and *E. vallentini*.**

| Prey | Fit | AICc | Delta AICc | AICc Wt | Cum Wt | $R^2$ | *p*-value |
|---|---|---|---|---|---|---|---|
| *Calanoides patagoniensis* | **Linear** | **237.42** | **0.00** | **0.93** | **0.93** | **0.85** | **<0.001** |
| *Calanoides patagoniensis* | Logarithmic | 242.74 | 5.33 | 0.07 | 1.00 | 0.81 | <0.001 |
| *Euphausia vallentini* | **Linear** | **108.37** | **0.00** | **0.79** | **0.79** | **0.68** | **<0.001** |
| *Euphausia vallentini* | Logarithmic | 111.08 | 2.71 | 0.21 | 1.00 | 0.64 | <0.001 |

**Note:**
The best fitting model according to second order Akaike Information Criterion (AICc) is bolded. The table also shows the difference in AICc between both models (Delta AICc), the relative weight of each model (AICc Wt), the cumulative weight of the models (Cum Wt), the proportion of the variance explained by each model ($R^2$) and model significance (*p*-value).

## Effect of prey abundance on *D. dianthus* ingestion rate

Coral ingestion rate showed a good linear fitting with the abundance of prey (Fig. 3) agreeing with the results expected for a functional response Type I (non-significant terms in Table 2). Model selection indicates that the linear fitting model is the most probable one for both prey items, explaining the 85% and 68% of the variance for *C. patagoniensis* and *E. vallentini*, respectively (Table 3).

## Effects of prey type on the daily ration of *D. dianthus*

The daily ration of *D. dianthus* feeding on *C. patagoniensis* and *E. vallentini* ranged from 0 to 8.19% and from 0 to 274.37% of coral carbon biomass $d^{-1}$, respectively (Fig. 4). Coral daily ration presented a similar response to the amount of carbon biomass offered despite of the different prey types supplied, that is, *C. patagoniensis* and *E. vallentini*

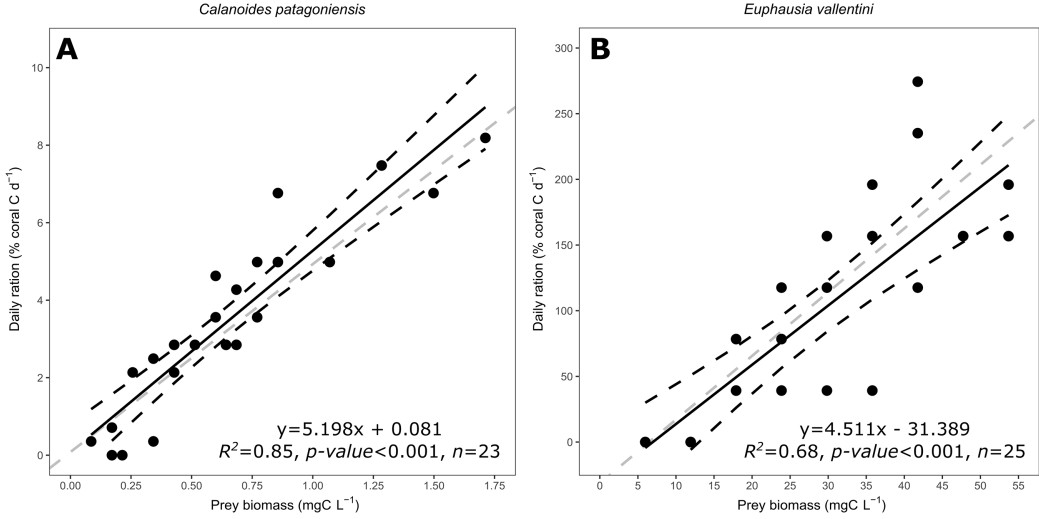

**Figure 4 Scatter plot showing the relationship between prey biomass (mg C L$^{-1}$) and coral daily ration (% of coral carbon biomass d$^{-1}$).** (A) Data for *C. patagoniensis*. (B) Data for *E. vallentini*. Black solid lines represent linear fits, whereas black dashed lines correspond to their 95% confidence intervals. Gray dashed lines correspond to a linear regression with a slope of 4.85.

(Fig. 4). According to ANCOVA results the slopes displayed by *D. dianthus* feeding on both prey items did not differ significantly ($F = 0.002$, $p$-value = 0.9685), which means that *D. dianthus* feeds similarly regardless of the prey type.

## DISCUSSION

The CWC *D. dianthus* was able to efficiently prey on medium and large sized zooplankton, even when corals faced prey abundances much higher than natural ones. To the best of our knowledge, this is the first direct assessment of *D. dianthus* feeding on naturally co-occurring zooplankton. *D. dianthus* displayed a functional response Type I, which implies that coral ingestion rate increases linearly with prey abundance. This response is typical for suspension feeders in general (*Jeschke, Kopp & Tollrian, 2004*) and corals in particular (*Anthony, 1999*), which passively extend their crown of nematocyst-laden tentacles and minimize handling by effectively paralyzing their prey, even at high densities. *D. dianthus* exhibited this response, that is, functional response Type I, when feeding on a medium-sized copepod (*C. patagoniensis*) and a large euphausiid (*E. vallentini*). Besides, the daily ration of corals increased similarly with the carbon biomass offered to corals regardless of the prey species, suggesting that *D. dianthus* is potentially able to feed on a wide variety of zooplankton prey.

### *D. dianthus* functional response feeding on zooplankton

The proportion of prey ingested by *D. dianthus* did not show any clear pattern with prey abundance (Fig. 2; Table 2), indicating a functional response Type I when corals feed on *C. patagoniensis* and *E. vallentini* (Fig. 3). Functional response Type I implies that prey handling time is so brief that it has a negligible effect on the coral food uptake, which leads to the high ingestion rates display by more effective predators (*Haddaway et al.,*

*2012*). *D. dianthus* is able to instantly paralyze even large krill (personal observation J. Höfer 2016), explaining its non-saturation state when facing very high abundances of krill.

Previous studies using stable isotopes suggested that *D. dianthus* may feed mainly on zooplankton instead of directly assimilate particulate organic matter, even presenting a slightly higher trophic level than planktivorous fishes (*Carlier et al., 2009*; *Mayr et al., 2011*). Laboratory studies also showed that *Desmophyllum* was able to capture *Artemia* (*Tsounis et al., 2010*), which sustained coral respiration, growth and organic matter release (*Naumann et al., 2011*). However, to this point, evidence of *D. dianthus* feeding effectively on naturally occurring zooplankton was lacking.

### Effect of prey abundance and prey type on *D. dianthus* feeding

The ingestion rate of corals feeding on *C. patagoniensis* and *E. vallentini* increased linearly with prey abundance (Fig. 3). During incubations corals experienced prey abundances (2.53–50.63 copepods $L^{-1}$ and 1.26–11.39 euphausiids $L^{-1}$) that were two to five orders of magnitude higher than the natural abundances registered for copepods (0.19 copepods $L^{-1}$) and euphausiids (0.0004 euphausiids $L^{-1}$) within the Comau Fjord (*Sánchez, González & Iriarte, 2011*). Despite these very large abundances, the feeding of *D. dianthus* displayed no saturation, suggesting that *D. dianthus* is able to feed effectively on dense zooplankton aggregations swimming near corals. Here, we recorded a maximum capture rate of 7.67 prey $polyp^{-1}$ $h^{-1}$ during *C. patagoniensis* incubations, agreeing with previous records (8.48 prey $polyp^{-1}$ $h^{-1}$) of *Desmophyllum* feeding on *Artemia* under optimal current conditions (*Tsounis et al., 2010*), which supports our findings that *D. dianthus* was effectively feeding under very high prey abundances. Previous studies have shown that zooplankton aggregations increase coral feeding (*Genin et al., 2005*) and zooplankton swarms are common in fjord boundaries (*Hirche, Laudien & Buchholz, 2016*) such as the steep slopes surrounding the inner part of the Comau Fjord (*Fillinger & Richter, 2013*). Therefore, being able to feed properly on moving zooplankton swarms might be an adaptive strategy for CWC living in a patchy and zooplankton scarce environment (*Kiriakoulakis et al., 2005*).

*D. dianthus* daily ration showed the same relationship with the carbon biomass offered regardless of the prey type, that is, slopes showed no significant difference in Fig. 4 (ANCOVA, $F = 0.002$, $p$-value = 0.9685). Additionally, *Desmophyllum* is able to effectively capture microzooplankton like nauplii (*Tsounis et al., 2010*), giving further support to *D. dianthus* as a skilled generalist zooplankton predator. *D. dianthus* was able to capture a slightly higher percentage of copepods (21.99%) than krill (18.22%), which is surprising since copepods are known to display an actual escape response to avoid predators (*Strickler & Bal, 1973*; *Yen et al., 1992*). Copepods escape from predators when they detect the hydromechanical disturbances caused by them (*Kerfoot, 1978*; *Haury, Kenyon & Brooks, 1980*). The magnitude of this signal (i.e., velocity difference between predator and the environment) determines the efficiency of predator detection and therefore copepod and nauplii escape success against filter feeders, swimming, and ambush predators (*Viitasalo et al., 1998*; *Kiørboe, Saiz & Visser, 1999*; *Green et al., 2003*; *Titelman & Kiørboe, 2003*).

However, CWC (including *D. dianthus*) passively extend their tentacles to feed, which probably makes them virtually undetectable for copepods and nauplii since the passively extended tentacles are very difficult to distinguish from the surrounding waters using hydromechanical cues. The feeding strategy of *D. dianthus* enables its ability to effectively capture copepods and nauplii, acting as a "ghost predator".

## General insights into *D. dianthus* feeding ecology

Cold-water corals were expected to not feed on zooplankton due to its paucity in the waters surrounding them (*Kiriakoulakis et al., 2005*). However, stable isotopes (*Duineveld, Lavaleye & Berghuis, 2004*; *Carlier et al., 2009*; *Mayr et al., 2011*), laboratory experiments (*Tsounis et al., 2010*; *Naumann et al., 2011*), and the present findings suggest that zooplankton is a major food source for CWC, including *D. dianthus*. Zooplankton behavior promotes the formation of dense aggregations or swarms (*Folt & Burns, 1999*; and references therein). Although these swarms may be spatially and temporally scattered for corals, our results suggest that *D. dianthus* is able to prey effectively on dense zooplankton aggregations such as krill swarms performing diel vertical migration. *E. vallentini* plays a major role in the pelagic food webs of the Chilean fjord region (*González et al., 2009*, *2010*, *2011*, *2016*), where *E. vallentini* migrates from 200 m depth to surface waters (*Hamame & Antezana, 2010*), representing an optimal feeding opportunity for *D. dianthus* according to our findings. These dense vertical migrating swarms of *E. vallentini* within the Comau Fjord might be fueling the high growth rates registered there (2.2–10 mm year$^{-1}$, *Jantzen et al., 2013b*) compared to other areas (0.5–2.2 mm year$^{-1}$, *Adkins et al., 2004*). Besides, this extra food supply may help *D. dianthus* to cope with the low pH (*McCulloch et al., 2012a*) that part of its population experiences within the Comau Fjord (*Jantzen et al., 2013a*; *Fillinger & Richter, 2013*).

Cold-water corals, such as *D. dianthus*, live in cooler waters than their tropical relatives, which implies less energy losses due to metabolic costs. Although zooplankton may be available only during short pulses, *D. dianthus* is able to seize these dense aggregations to feed actively as it is shown by the present and previous findings (*Tsounis et al., 2010*). These short "feeding windows" may be enough for *D. dianthus* to avoid starvation and grow, while, between these "feeding windows" corals might exploit particulate (detritus, nano- and microplankton) and/or dissolved organic matter as it has been recorded for other corals (*Orejas et al., 2001*; *Orejas, Gili & Arntz, 2003*; *Ribes, Coma & Rossi, 2003*; *Tsounis et al., 2006*).

## CONCLUSIONS

The ingestion rate of *D. dianthus* increased linearly with prey abundance (i.e., functional response Type I) even when corals experienced prey abundances that were much higher than the natural ones. This implies that *D. dianthus* is a capable zooplankton predator that seems to be adapted to exploit dense zooplankton aggregations when they pass by. Finally, *D. dianthus*, feeding response showed no differences when preying upon a medium-sized copepod or a large euphausiid, which evidences, along with previous studies, that this CWC is able to effectively feed on a wide variety of zooplankton prey.

## ACKNOWLEDGEMENTS

We are very grateful to the staff of Huinay Scientific Field Station, especially G. Försterra, for their logistical support. F. Beaujot, A. Thomasberger and M. Schiønning assisted collecting corals, while P. Martis and N. García-Herrera helped during incubations. This is a contribution of the IDEAL research center and publication number 160 with contribution from Huinay Scientific Field Station. N. García-Herrera kindly provided the pictures for Fig. 1. Useful comments from the editor and two reviewers helped us to improve the manuscript.

### Funding

This work was supported by the bilateral Chilean-German PACOC Project (CONICYT-BMBF 20140041; BMBF 01DN15024) as well as CONICYT FONDAP-IDEAL 15150003 and AWI (PACES II, Topic 1, WP6). The funders had no role in study design, data collection and analysis, decision to publish, or preparation of the manuscript.

### Grant Disclosures

The following grant information was disclosed by the authors:
Bilateral Chilean-German PACOC Project: CONICYT-BMBF 20140041; BMBF 01DN15024.
CONICYT FONDAP-IDEAL 15150003 and AWI: PACES II, Topic 1, WP6.

### Competing Interests

The authors declare that they have no competing interests.

### Author Contributions

- Juan Höfer conceived and designed the experiments, performed the experiments, analyzed the data, contributed reagents/materials/analysis tools, prepared figures and/or tables, authored or reviewed drafts of the paper, approved the final draft.
- Humberto E. González conceived and designed the experiments, contributed reagents/materials/analysis tools, authored or reviewed drafts of the paper, approved the final draft.
- Jürgen Laudien conceived and designed the experiments, contributed reagents/materials/analysis tools, authored or reviewed drafts of the paper, approved the final draft.
- Gertraud M. Schmidt contributed reagents/materials/analysis tools, authored or reviewed drafts of the paper, approved the final draft.
- Verena Häussermann contributed reagents/materials/analysis tools, authored or reviewed drafts of the paper, approved the final draft.
- Claudio Richter conceived and designed the experiments, contributed reagents/materials/analysis tools, authored or reviewed drafts of the paper, approved the final draft.

## Field Study Permissions

The following information was supplied relating to field study approvals (i.e., approving body and any reference numbers):

The collection of animals for scientific purposes was approved by the sub-secretariat of fisheries and farming within the Chilean Ministry of Economy, Development & Tourism (file number: 1760).

## Data Availability

Höfer, Juan; Garcia Herrera, Nur; Martis, Paula; Laudien, Jürgen; González Estay, Humberto; Richter, Claudio (2018): Functional response of the cold-water coral *Desmophyllum dianthus* feeding in vitro on krill and copepods. Alfred Wegener Institute, Helmholtz Center for Polar and Marine Research, Bremerhaven, PANGAEA, https://doi.pangaea.de/10.1594/PANGAEA.893159.

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
