# Peer review of "All you can eat: the functional response of the cold-water coral Desmophyllum dianthus feeding on krill and copepods"

_PeerJ, doi:10.7717/peerj.5872_

## Round 0.1 · original submission · Minor Revisions

I have heard back from two reviewers, both of whom have offered constructive comments to help you improve your manuscript. I have gone over the manuscript and comments myself, and agree with the comments, while also noting that your work is well presented. Thus, my decision is minor revisions, and I do not think you will need to expend too much effort to bring it up to a publishable standard. I look forward to seeing a revised version of your work.

·

Basic reporting

The manuscript is well written and clear, and professional English have been used throughout.
Literature references have been well selected and they provide good background and context for the manuscript.
The manuscript has a good structure and all figs/tabs/supplementary material have been well cited and provided. Nevertheless, Video S1 has low resolution and Figure and Video captions of supplementary material are missing.
Hypotheses are well expressed and results are well presented and relevant to the hypotheses

Experimental design

The manuscript is an original primary research and reflect the aims and scope of the journal.
Research questions are relevant and well defined. Authors have clearly identified the knowledge gaps and the method to develop an experimental design able to answer the question and fill the scientific gaps.
The research investigation has been performed with good technical & ethical standard.
Method and materials are well described with good level of details and information. However, in order to facilitate the understanding to the reader, further clarifications are needed:
Line 131. In the incubation experiments is not clear if corals are in the glass bottles with one of the prey type and in two different days as suggested in table 1. See my comments in table 1
Line 142. Back at HSFS Lab, It is not clear an aspect of the experiment. At the beginning of the section/paragraph it should be said the samples are moved from HSFS lab- acclimation tanks to HSFS jetty- in situ experiment.
TABLE 1
If date is the incubation date, following the protocol explained in the test it means that e.g :
09/09 dive day with 20 coral specimens
10-11/09 acclimation days (48h)
11/09 incubation day
If this is correct, you should have (in theoy) in the table"
11/09 EV N 10
11/09 CP N 10

Since there is a difference between the theory and the info in the table, you should explain (at least in a note in the table)
- why are there two different days for zooplankton (one for ev and one for cp) . Couldn't be same day with different bottles for each prey type?
- different number of coral specimens than expected and for each prey type
- there are repeats for each experiment (same month, same prey type, same day) but not replicates. Is there any reason why replicates have been not considered in the experiment design?

Validity of the findings

The novelty of the results will provide good impact to the scientific knowledge. Data are robust and statistically sound providing conclusive results.
Conclusion are well stated, linked to original research question & limited to supporting results.

Additional comments

INTRO
Line 47. Taxonomic nomenclature rule is not applied. It should be Desmophyllum dianthus (Esper, 1794)
Line 51. Taxonomic nomenclature rules is not applied. It should be Desmophyllum pertusum (syn. Lophelia pertusa).
See WoRMS update, L. pertusa is not accepted anymore
http://www.marinespecies.org/aphia.php?p=taxdetails&id=1245747

New combination reference Addamo et al (2016)
Addamo, A. M.; Vertino, A.; Stolarski, J.; García-Jiménez, R.; Taviani, M.; Machordom, A. (2016). Merging scleractinian genera: the overwhelming genetic similarity between solitary Desmophyllum and colonial Lophelia. BMC Evolutionary Biology. 16(1)., available online at https://doi.org/10.1186/s12862-016-0654-8

Line 53. It is not clear why the sentence starts with "while”. It should be shallow coral banks
Line 55. Shrink to deeper or shallower communities?
Line 57. Please use the same format for geo-coordinates should be used, see line 99
Line 62. Space should be included 7 m

Line 87. Taxonomic nomenclature rule is not applied. It should be Calanoides patagoniensis (Brady, 1880)
Line 88. Taxonomic nomenclature rule is not applied. It should be Euphausia vallentini Stebbing, 1900

MAT&MET

Line 93. It should be Desmophyllum dianthus
Line 96. Capital letter for official name of ministry, it should be Ministry of Economy, Development and Tourism
Line 97. Case means season or month or dive/survey? It needs to be clarified
Line 97. It should be circa or max twenty specimens. See my comments in table 1
Line 99. Please use the same format for geo-coordinates see line 57
Line 132. It should be Desmophyllum dianthus
Line 135. I have a doubt about Turbulence? if the corals are inside glass bottles, the ins situ conditions of turbulence is not respected the real condition, because they are affected by glass bottle, which works as closed cage. Isn't it?
Line 143. The remaining zooplankton that have been not captured by corals (?). it will be helpful clarify it.
Line 157. Month or season?

DISCUSSION
Line 230. Just an observation: since corals has been moved from their natural substrate, to lab, and finally to jetty, shouldn’t be considered as semi-naturally or slightly manipulated naturally conditions?
Line 231. It should be Desmophyllum dianthus
Line 235. It should be Desmophyllum dianthus
Line 245. It should be Desmophyllum dianthus
Line 272. It should be Desmophyllum dianthus
Line 276. It should be Desmophyllum dianthus
Line 297. It should be Euphausia vallentini

Reviewer 2 ·

Basic reporting

The authors make a convincing case that Desmophyllum dianthus is a generalist zooplankton predator capable of exploiting dense aggregations of zooplankton over a wide prey size-range – i.e., a type I functional response. This does not preclude that the species may also employ the food sources of particulate or dissolved organics. Their experimental methodology appears to be sound and they have exploited this species at an upwelling location, which makes it more accessible than most other deep-water coral species. I appreciate the allusion of D. dianthus as a “ghost predator”, a fitting analogy. Feeding behavior is not my specialty, but it seems to me that the authors have made their point.
Let me just make a few small remarks:
I think the plural of prey is prey, not preys. I may be wrong.
Line 56: Why would global warming affect deep-water corals?
Supplemental video 1 is of zooplankton swimming. What is the purpose of this video. I don’t think it is referenced in the text.
Line 307: replace implicates to implies.

Experimental design

See above

Validity of the findings

See above.

Additional comments

The authors make a convincing case that Desmophyllum dianthus is a generalist zooplankton predator capable of exploiting dense aggregations of zooplankton over a wide prey size-range – i.e., a type I functional response. This does not preclude that the species may also employ the food sources of particulate or dissolved organics. Their experimental methodology appears to be sound and they have exploited this species at an upwelling location, which makes it more accessible than most other deep-water coral species. I appreciate the allusion of D. dianthus as a “ghost predator”, a fitting analogy. Feeding behavior is not my specialty, but it seems to me that the authors have made their point.
Let me just make a few small remarks:
I think the plural of prey is prey, not preys. I may be wrong.
Line 56: Why would global warming affect deep-water corals?
Supplemental video 1 is of zooplankton swimming. What is the purpose of this video. I don’t think it is referenced in the text.
Line 307: replace implicates to implies.

---

## Round 0.2 · Minor Revisions

As pointed out by one of the reviewers in the previous round, the authorities of species are often incorrectly formatted or have the incorrect year (e.g. Esper, 1974). Note that authorities are NOT references in their formatting, and please add the authority for all species upon first mention in the text. You can check the formatting and correctness of each one via WoRMS.

Other than this, the paper is well-revised, and I anticipate accepting this soon upon receiving a corrected version.

---

## Round 0.3 · accepted · Accept

I am happy to move this into production; it has been well-revised and all outstanding issues have been dealt with. Congratulations!

#